# Native mass spectrometry and structural studies reveal modulation of MsbA–nucleotide interactions by lipids

Tianqi Zhang[1], Jixing Lyu[1], Bowei Yang[2], Sangho D. Yun[1], Elena Scott[1], Minglei Zhao ®[2] & Arthur Laganowsky ®[1]✉

The ATP-binding cassette (ABC) transporter, MsbA, plays a pivotal role in lipopolysaccharide (LPS) biogenesis by facilitating the transport of the LPS precursor lipooligosaccharide (LOS) from the cytoplasmic to the periplasmic leaflet of the inner membrane. Despite multiple studies shedding light on MsbA, the role of lipids in modulating MsbA-nucleotide interactions remains poorly understood. Here we use native mass spectrometry (MS) to investigate and resolve nucleotide and lipid binding to MsbA, demonstrating that the transporter has a higher affinity for adenosine 5'-diphosphate (ADP). Moreover, native MS shows the LPS-precursor 3-deoxy-D-*manno*-oct-2-ulosonic acid $(Kdo)_2$-lipid A (KDL) can tune the selectivity of MsbA for adenosine 5'-triphosphate (ATP) over ADP. Guided by these studies, four open, inward-facing structures of MsbA are determined that vary in their openness. We also report a 2.7 Å-resolution structure of MsbA in an open, outward-facing conformation that is not only bound to KDL at the exterior site, but with the nucleotide binding domains (NBDs) adopting a distinct nucleotide-free structure. The results obtained from this study offer valuable insight and snapshots of MsbA during the transport cycle.

Gram-negative bacteria, including *E. coli*, possess a complex envelope consisting of an inner membrane and an outer membrane separated by the periplasm[1–3]. The inner membrane forms a typical phospholipid bilayer surrounding the cytoplasm of bacteria, while the outer membrane adopts an asymmetric structure, with phospholipids comprising the inner leaflet and lipopolysaccharides (LPS) as the major component of the outer leaflet[1–3]. LPS plays a vital role in maintaining an effective outer membrane barrier, providing resistance against antibiotics and various environmental stresses[4,5]. Notably, MsbA, a member of the ATP-binding cassette superfamily, plays a crucial role in LPS biosynthesis by facilitating the flipping of the LPS-precursor lipooligosaccharide (LOS) from the cytoplasmic side of the inner membrane to the periplasmic side[6,7]. The essentiality of *E. coli* MsbA is evident from studies reporting that MsbA knockouts are lethal[7,8], making this transporter an attractive target for developing antibiotics that inhibit function thereby combating multidrug-resistant infections.

Numerous investigations have shed light on the function, structure, and mechanism of MsbA[9–17]. MsbA forms a homodimer and exhibits a topology similar to other ABC transporters, comprising two NBDs and two transmembrane domains (TMDs) containing 12 transmembrane helices[18,19]. More specifically, the NBDs contain a RecA-type ATP binding core ($RecA_{core}$), composed of six beta sheets and four alpha helices (helices A-D), that is decorated with three additional beta sheets and an alpha-helical subdomain (ABCα)[20,21]. Other conserved motifs of the NBDs of MsbA include the hydrophobic residue of the A-loop (residue 351), Walker A or P-loop (GxxGxGK(S/T), where x denotes any residue, resides 376-383), Q-loop (φ(φ/Q)Q, where φ denotes a hydrophobic residue, residues 421-424), X-loop (a feature characteristic of ABC exporters, residues 472-478), C-loop (formed by

[1]Department of Chemistry, Texas A&M University, College Station, TX, USA. [2]Department of Biochemistry and Molecular Biology, University of Chicago, Chicago, IL, USA. ✉e-mail: ALaganowsky@chem.tamu.edu

the signature motif with consensus sequence of LSGGQ, residues 481-487), Walker B ($\phi_4$D, residues 501-506), D-loop (consensus sequence of SALD, residues 509-512), and the H-switch histidine (residue 537)[20–22].

A 'trap and flip' model has been put forth to explain MsbA-mediated LPS transport[10,13]. In the absence of nucleotides, MsbA adopts an open, inward-facing (IF) conformation with separated NBDs, which is thought to facilitate the entry of cytoplasmic LOS[23]. Upon binding ATP, MsbA undergoes dimerization of the NBDs, inducing rearrangement in the TMDs. An ATP hydrolysis-driven conformational change promotes the transition to an outward-facing (OF) conformation, facilitating the flipping of LOS to the periplasmic side of the inner membrane for further modifications. Following inorganic phosphate release, MsbA returns to an IF conformation. Although the proposed mechanism is widely accepted, additional evidence is needed to better understand the intricate details of the transport cycle, particularly concerning the interactions of MsbA with nucleotide and lipids during transport.

Similar to other ABC transporters, the ATPase activity of MsbA is stimulated by various substrates[24,25]. One of the known hexaacylated lipid A substrates is KDL, a molecule consisting of a lipid A core modified with 3-deoxy-D-*manno*-oct-2-ulosonic acid (Kdo) disaccharide[24–27]. Recent studies have identified binding sites for LPS precursors on MsbA that are important for stimulation of the transporter[10,11,13,17]. One binding site resides within the inner cavity of the protein where several basic residues coordinate the headgroup of LOS. More recent findings have identified an exterior binding site on MsbA, which was revealed in a structure of the transporter trapped in an open, outward-facing conformation[11]. Mutations introduced to disrupt binding at either binding site abolish or reduce lipid-induced stimulation of ATPase activity[10,11].

Native mass spectrometry, or non-denaturing mass spectrometry, is distinctly positioned to study the interactions between membrane proteins and small molecules, such as lipids[28–30]. One of the strengths of the technique lies in the ability to preserve noncovalent interactions and native-like structures of membrane proteins within the mass spectrometer, enabling the examination of individual ligand-binding events to protein complexes[31,32]. This technique has been extensively employed to unravel vital information, ranging from membrane protein-soluble protein interactions, membrane protein-lipid interactions, as well as interactions between proteins and other molecules, including metals and drugs[33–45]. Furthermore, by utilizing native MS in conjunction with a temperature controller, it is possible to determine thermodynamic parameters for protein-protein and protein-ligand interactions[37,46–50], which is important in understanding the molecular forces that drive non-covalent interactions[51].

In this work, we characterize MsbA-nucleotide interactions and show how these interactions can be influenced by lipids. Native MS results capture MsbA hydrolyzing ATP and illuminate different nucleotide-binding states. In the presence of specific lipids, MsbA populates distinct lipid and nucleotide-bound states. Structural studies of MsbA under similar conditions lead to the determination of five structures, one of which is bound to KDL with the NBDs populating a distinct, nucleotide-free conformation. These results provide additional insight into how lipids modulate MsbA-nucleotide interactions.

## Results

### Determination of MsbA-nucleotide equilibrium dissociation constants

The role of MsbA in binding and hydrolyzing ATP to fuel the transport cycle motivated us to investigate and determine the equilibrium dissociation constants for ATP and ADP (Fig. 1 and Supplementary Fig. 1). In our previous work, we optimized samples of *E. coli* MsbA solubilized in the $C_{10}E_5$ (pentaethylene glycol monodecyl ether) detergent to ensure that small molecule binding to the transporter can be resolved,

including copper(II) binding to the N-terminus of MsbA[11]. These samples provide the opportunity to probe nucleotide binding to MsbA. As the N-terminus of MsbA binds copper(II), and metal binding can influence lipid binding, the transporter was incubated with copper(II) ions prior to buffer exchange to saturate the N-terminal metal binding sites. MsbA was mixed with 20 µM ATP and 10 µM $Mg^{2+}$ and immediately injected into the mass spectrometer to minimize any ATP hydrolysis. The mass spectrum showed signals for the binding of one and two ATP molecules to homodimeric MsbA (Fig. 1a, b). To determine the equilibrium dissociation constants ($K_{Dn}$) for binding of the $n$th nucleotide, MsbA underwent titration with varying concentrations of ATP. Deconvolution of the mass spectra[52] enabled the determination of the mole fraction of MsbA(ATP)$_{0-2}$ (Fig. 1e). Following a similar approach to our previous study[41], we applied a sequential ligand binding model to deduce $K_D$s. The results revealed that $K_{D1}$ is 47.8 ± 2.5 µM and $K_{D2}$ is 124.4 ± 6.2 µM (Fig. 1g). Analogous experiments were performed to determine $K_D$s for ADP binding. MsbA in the presence of 20 µM ADP and 10 µM $Mg^{2+}$ displayed a higher abundance of MsbA binding one and two ADPs (Fig. 1c, d). The $K_D$s for binding the first and second ADP were 17.8 ± 1.3 µM and 62.3 ± 4.7 µM, respectively (Fig. 1f, g).

Taken together, the nucleotide-binding data indicates MsbA binds ADP with higher affinity than ATP. This observation is consistent with an earlier report that reported estimates of $K_D$ for ATP and ADP binding of 3050 and 130 µM, respectively[26]. While the overall trend is consistent, it is important to note that native MS measurements enable the resolution of each nucleotide binding to MsbA, which is important for determining $K_{Dn}$ and providing more meaningful values. We initially kept the concentration of $Mg^{2+}$ at 10 µM in the nucleotide-binding experiments to minimize adduction of the cation, which can hinder the quality of mass spectra when $Mg^{2+}$ is at a much higher concentration (Supplementary Fig. 2). However, we also determined $K_D$s for ATP and ADP binding to MsbA in the presence of excess $Mg^{2+}$ relative to the concentration of nucleotide (Supplementary Fig. 3). No statistical difference in $K_D$s was found (Supplementary Table 2) but we did observe at a higher concentration of $Mg^{2+}$ the ATPase activity of MsbA was enhanced.

### Monitoring ADP and lipid binding to MsbA

Prior research has demonstrated that MsbA selectively binds lipids, with the two tightest binding lipids identified as TOCDL (1,1′,2,2′-tetraoleoyl-cardiolipin) and KDL[11]. Focusing on these two lipids, we set out to determine their impact on ADP binding to MsbA (Fig. 2 and Supplementary Fig. 4). The mass spectrum of MsbA in the presence of ADP and $Mg^{2+}$ but with no lipid showed a strong signal for the binding of one and two ADP molecules (Fig. 1). The addition of 3 µM TOCDL to the sample resulted in the appearance of up to four TOCDLs bound to MsbA (Fig. 2a). The abundance of one and two ADP molecules for each of the TOCDL bound states did not significantly change, implying that ADP and TOCDL binding are independent. Upon incubating the sample for a period of time, there was no change in the abundance of TOCDL and ADP, as expected, since MsbA does not hydrolyze ADP (Supplementary Fig. 5a). In the case of KDL (1 µM final), and with the same concentration of ADP and $Mg^{2+}$, the binding of one to three KDL molecules to MsbA was observed. There was a slight variation in the abundance of one and two ADP molecules between the unbound and KDL bound states of MsbA (Fig. 2b). More specifically, MsbA not bound to KDL showed a higher abundance of the nucleotide-free state compared to the two ADP bound state. In contrast, the binding of three KDLs enhanced the abundance of the two ADP-bound states. Like TOCDL, incubation of the sample did not alter the abundance of KDL and ADP bound to MsbA (Supplementary Fig. 5b). The mass spectra of MsbA in the presence of the same amount of TOCDL/KDL but with no ADP and $Mg^{2+}$ (Fig. 2c, d), showed the binding of ADP slightly enhanced KDL binding, as evident by the increase in abundance of the lipid-

 

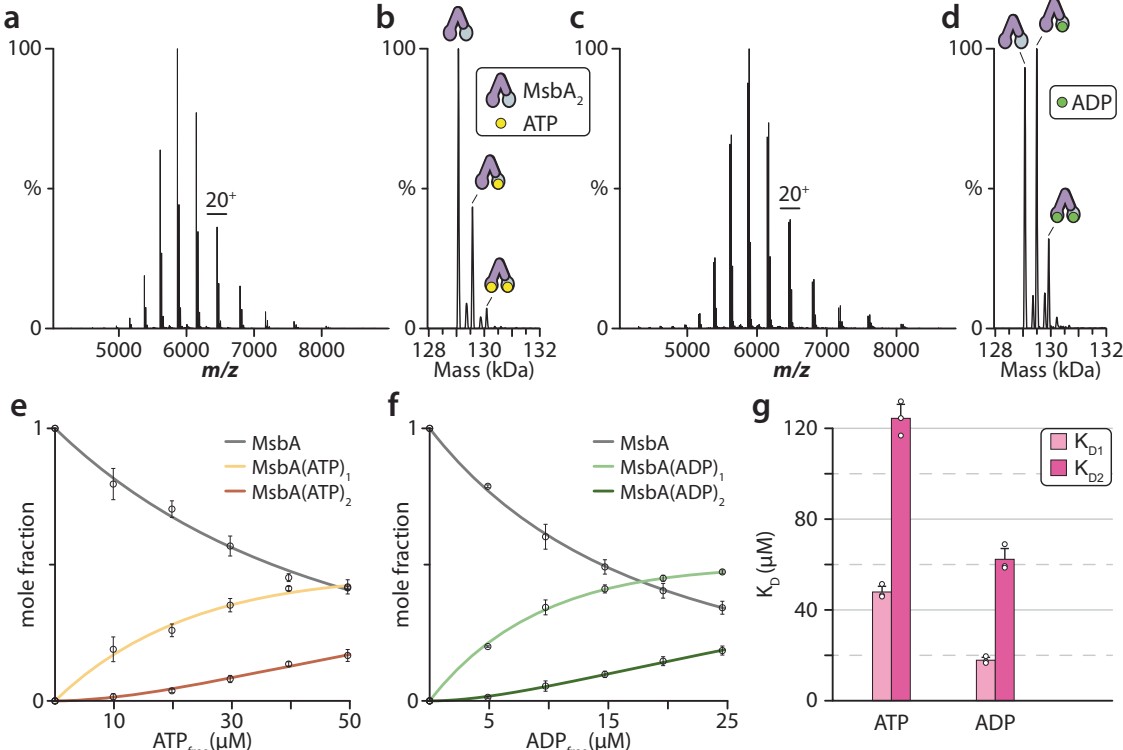

**Fig. 1 | Determination of equilibrium dissociation constants ($K_D$) for individual nucleotide-binding events to MsbA. a** Representative native mass spectrum of MsbA (0.5 μM) in the presence of 10 μM $Mg^{2+}$ and 20 μM ATP. **b** Deconvolution of the mass spectrum shown in panel a. **c** Representative native mass spectrum of MsbA (0.5 μM) in the presence of 10 μM $Mg^{2+}$ and 20 μM ADP. **d** Deconvolution of the mass spectrum shown in (**c**). **e** Plot of mole fraction data for MsbA(ATP)$_{0-2}$ determined from the titration series (dots) and resulting fit from a sequential ligand-binding model (solid lines). **f** Plot of mole fraction data for MsbA(ADP)$_{0-2}$ determined as described for (**e**). **g** $K_{Dn}$ values for the $n$th nucleotide binding to MsbA. Reported are the mean and standard deviation ($n = 3$, biological replicates). Source data are provided as a Source Data file.

bound states. In short, the results demonstrate that lipids can influence MsbA-ADP interactions.

**Probing lipid and ATP binding to MsbA.** We then investigated lipid and ATP binding to MsbA, which is more complex due to the hydrolysis of ATP, leading to a more dynamic environment (Fig. 3 and Supplementary Fig. 6). We opted to keep samples cold to monitor turnover and slow down the reaction (the activity at 10 °C is ~5% of that at 37 °C)[25], enhancing the likelihood of uncovering additional details of the MsbA transport cycle. Like the studies determining ATP binding affinity, introducing MsbA into the mass spectrometer immediately after adding ATP and $Mg^{2+}$ showed binding of one and two ATP molecules with no evidence of ATP hydrolysis (Fig. 3a). After a 30-minute incubation, additional peaks are present that correspond to MsbA bound to either ATP or ADP, two ATPs, and mixed nucleotides, specifically MsbA(ATP)(ADP) (Fig. 3a). These results suggest that the hydrolysis of two ATP molecules, each bound to a separate NBD, does not necessarily occur simultaneously, i.e., fire independently.

To better understand how lipids impact the ATPase activity of MsbA, we performed similar experiments but in the presence of different lipids. The addition of TOCDL resulted in similar results observed for the transporter with ADP (Fig. 3b). The mass spectrum reveals the binding of up to four TOCDLs. The abundance of one and two ATPs was similar across the different TOCDL-bound states. Upon incubation, ATP is turned over along with the appearance of ADP binding (Fig. 3b). A noticeable distinction emerged when KDL is present instead of TOCDL (Fig. 3c). At the earliest time point, the presence of one or more KDL molecules bound to MsbA notably enhanced the binding of ATP, particularly when the transporter bound three to four KDL molecules. After 0.5 h of incubation, the

appearance of specific peaks with two ATPs bound became more pronounced (Fig. 3c). The higher KDL bound states were predominantly bound to ATP, implying that KDL binding enhances the specificity of MsbA for ATP over ADP.

Subsequently, we questioned whether other lipids present in the bacterial membrane could also impact MsbA-ATP interactions. We elected to examine the effects of various lipids found in the bacterial membrane, including phosphatidic acid (PA), phosphatidylcholine (PC), phosphatidylethanolamine (PE), phosphatidylglycerol (PG), and phosphatidylserine (PS), all containing the acyl chain composition 1-palmitoyl-2-oleoyl (PO, 16:0–18:1). It is worth noting that, with the exception of PC, these lipids are naturally present in *E. coli*[53,54]. For all of these lipids there was no significant change in the binding patterns for ATP or lipid when incubated together with MsbA (Supplementary Figs. 7–11), indicating that KDL specifically enhances the affinity of MsbA for ATP.

**Monitoring MsbA turnover in the presence of KDL.** Given the intriguing findings observed regarding MsbA-ATP interactions in the presence of KDL, we conducted a time course experiment spanning 10 h, recording mass spectra at 2-h intervals (Fig. 4). The sample was incubated on ice to slow down the reaction. For this time course, the binding of up to four KDL molecules was observed. The binding of one and two ATP molecules was also observed along with the appearance of ADP bound states at longer incubation time points. Throughout the time course we observed variation in the abundance of the two ATP-bound states of MsbA that is dependent on the number of KDL bound. More specifically, a plot of the mole fraction of MsbA(ATP)$_2$(KDL)$_{0-4}$ illustrates this variation (Fig. 4f and Supplementary Table 1). As MsbA turns over ATP, the abundance of peaks corresponding to proteins

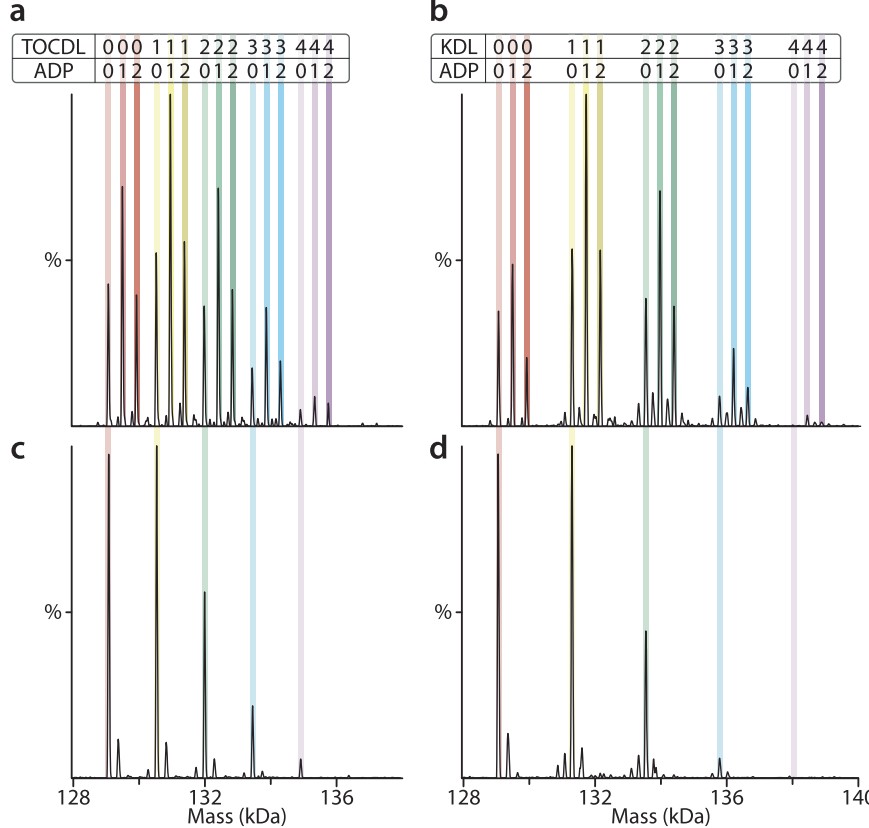

**Fig. 2 | Biophysical characterization of ADP and lipid binding to MsbA.**
**a** Deconvoluted mass spectrum is shown for 0.5 µM MsbA mixed with 10 µM Mg²⁺, 25 µM ADP and 3 µM TOCDL. Data was recorded right after mixing. **b** Deconvoluted mass spectrum of 0.5 µM MsbA mixed with 10 µM Mg²⁺, 25 µM ADP and 1 µM KDL. Data was acquired right after mixing. **c** Data for 0.5 µM MsbA mixed with 3 µM TOCDL. **d** Data for 0.5 µM MsbA mixed with 1 µM KDL. The number of lipids and ADPs bound are labeled.

bound with three or four KDLs increased, peaking at 6 h, while those bound with less KDL also did but with a slower rate. After four hours, ADP binding became more evident, indicating an appreciable amount of ATP had been hydrolyzed by MsbA. Again, ADP binding is most pronounced for MsbA(KDL)$_{0-1}$. For the larger number of KDLs bound to MsbA, the binding of ADP is not observed and MsbA preferentially binds ATP. Together, these results indicate that there are distinct stoichiometries populated in the transport cycle of MsbA.

**CryoEM structures of MsbA under turnover conditions.** The outcomes obtained from the native MS experiments inspired us to conduct cryoEM experiments under comparable conditions. MsbA (78 µM) solubilized in $C_{10}E_5$ was prepared in the presence of 2.5-fold molar excess of KDL (194 µM) and incubated in the presence of 1 mM ATP and Mg²⁺. We were able to obtain 3D reconstructions for a total of five structures, all with C2 symmetry imposed. Four of the structures, ranging from 3.6 to 3.9 Å resolution (Supplementary Table 4 and Supplementary Fig. 14), adopt open, inward-facing conformations (OIF) that vary in their degree of openness (Fig. 5). These open structures combined with three previously reported structures of MsbA[11,12,17] can be ranked in terms of NBD separation (Supplementary Table 3). One of the structures (OIF4, PDB 8TSR) displays the largest separation of 91.7 Å (Cα to Cα of T561). The second largest is OIF3 (PDB 8TSS) with an NBD distance of 89.9 Å followed by PDB 3B5W (85.1 Å). The third smallest distance of 79.3 Å corresponds to a structure we previously reported (PDB 8DMO). One of the IF structures (OIF2, PDB 8TSQ) is like a previously reported structure of MsbA from *Salmonella typhimurium* (PDB 6BL6), where the NBDs are separated by 75.9 Å. The fourth open structure (OIF1, PDB 8TSP) has the shortest separation (64.8 Å). These

structures provide snapshots of OIF conformations of MsbA and illustrate the dynamics of the transporter.

The fifth MsbA structure adopts an open, outward-facing conformation and was resolved to a resolution of 2.7 Å (Fig. 6). There are several notable features of this structure. First, well-resolved density is observed for KDL binding, even for the six lipid tails (Fig. 6a, b and Supplementary Fig. 15). The positioning of KDL is centered on TM5, occupying a specific region within the cytoplasmic leaflet of the inner membrane. This location has recently been identified as the exterior LOS binding site of MsbA, a site that is important for stimulation of MsbA ATPase by hexaacylated lipid A species[11]. The coordination of KDL is reminiscent of our recent structure with R188 and R238 interacting with the characteristic phosphoglucosamine (P-GlcN) moieties of LOS. However, an additional contact is formed between N235 and one of the Kdo molecules of the headgroup. While the structure is in an open, outward conformation, the transporter is not bound to nucleotide. This makes the structure rather unusual as MsbA and other ABC transporters are bound to nucleotide. To rule out the possibility that all the ATP had been consumed prior to freezing samples, we monitored ATPase activity assay under the same conditions used for cryoEM and found the transporter was still active, turning over ATP even 6 h after the time point we froze samples (Supplementary Fig. 12).

Another defining feature of the open, outward-facing structure is the conformation of the NBDs (Fig. 6c–e). Relative to the vanadate-trapped MsbA structure in an open, outward conformation (PDB 8DMM), parts of MsbA undergo rotation and translation, including the NBDs. This can be illustrated by aligning both structures, leading to superposition of the TMD and the RecA$_{core}$ (Fig. 6c). Specifically, the TMD aligns well apart from coupling helix 2, which is translated

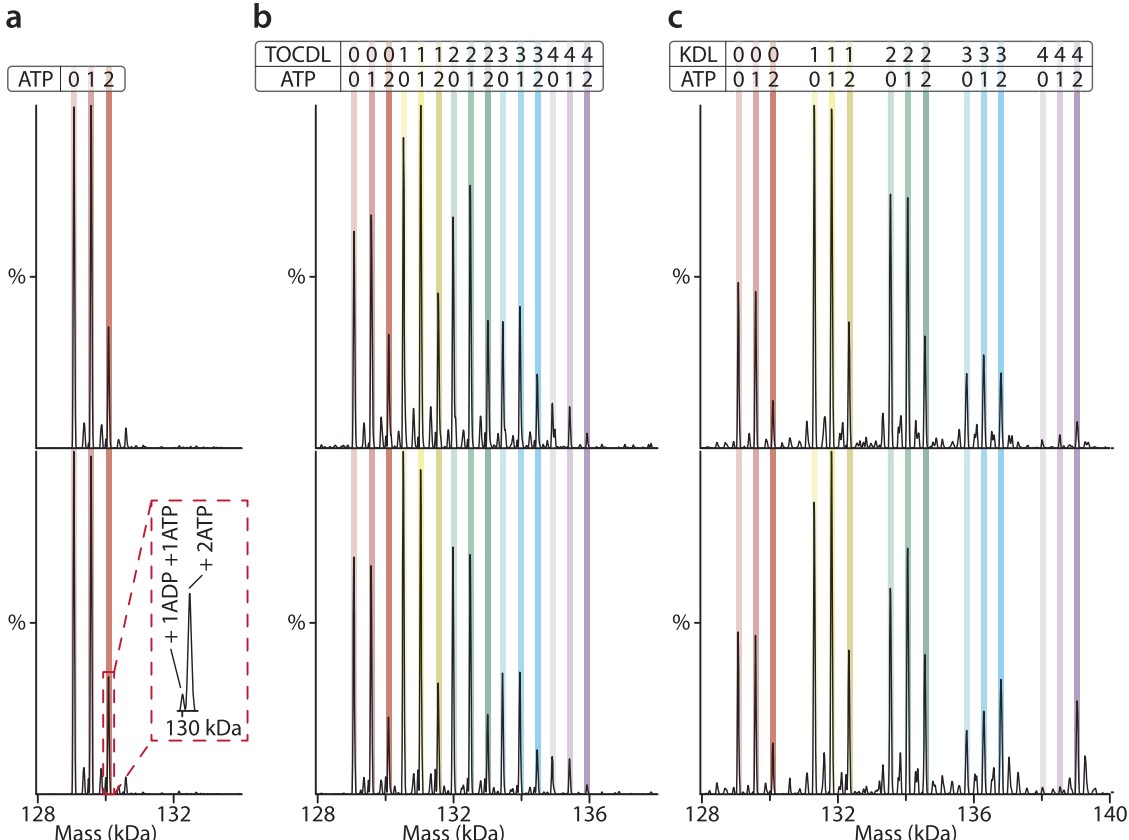

**Fig. 3 | Lipids modulate MsbA nucleotide interactions and populate distinct stoichiometries. a** Deconvoluted mass spectra of 0.5 μM MsbA mixed with 10 μM Mg$^{2+}$ and 50 μM ATP. Data were acquired after mixing (top) and incubation for 30 min (bottom). The peaks corresponding to MsbA binding ATP and ADP are labeled in zoomed inset. **b** Deconvoluted mass spectra of 0.5 μM MsbA mixed with

10 μM Mg$^{2+}$, 50 μM ATP and 3 μM TOCDL after mixing (top) and 30-min incubation (bottom). **c** Deconvoluted mass spectra of 0.5 μM MsbA mixed with 10 μM Mg$^{2+}$, 50 μM ATP and 1 μM KDL after mixing (top) and 30-min incubation (bottom). The number of lipids and ATPs bound are labeled.

upward by ~2.5 Å leading to slight distortion of TM4 and TM5 (Fig. 6c). Helix C (resides 513-526) of the NBD rotates ~21° away from the central twofold symmetry axis. The ABCα domain mirrors the rotation and direction of Helix B (residue 483-496) but to a lesser extent, albeit nearly 12°. These movements within the NBD lead to rearrangement altering the interface and nucleotide-binding pocket (Fig. 6d). The NBD interface is stabilized by interaction between T508 (positioned about the two-fold symmetry axis), R377′ of the P-loop interacts with the sidechain of E516 that is also stabilized by interaction with R538, and the backbone carbonyl of A510 (within the Walker B motif) with amide of S378′ (within the P-loop) (Fig. 6e). The rotation and translation of Helices B-C and ABCα domain opens the nucleotide binding pocket, where these components formed contacts with ADP and vanadate in the vanadate-trapped MsbA structure in an open, outward conformation (PDB 8DMM) (Fig. 6d). The π–π interaction of Y351 of the A-loop with the adenosine moiety (observed in nucleotide bound structures) is substituted by a cation–π interaction with R354 (Fig. 6f), priming the NBDs for binding ATP. This interaction suggests that the absence of nucleotide is not a result of the freezing process, and native MS data also shows the presence of nucleotide-free MsbA bound to KDL (Fig. 4a–c).

## Discussion

Through biophysical characterization of individual nucleotide-binding events to MsbA, it has been observed that there is a disparity in its affinity for ATP and ADP. The reported K$_D$ values differ from those previously reported[25,55]. The main reason for this difference lies in the

fact that previous measurements were ensemble averaged, i.e., unable to discern the binding of one versus two nucleotides. In contrast, native MS enables the biophysical characterization of one nucleotide at a time, thereby enabling more precise determination of binding affinities. Notably, MsbA exhibits a higher binding affinity for ADP compared to ATP. Native MS results for MsbA turning over ATP reveal binding of one ADP and one ATP, suggesting that hydrolysis of the two ATPs bound to MsbA does not happen simultaneously.

An interesting observation is the role of lipid in selectively modulating the affinity of MsbA for nucleotides. For example, in the presence of KDL, ATP binding affinity is significantly enhanced, leading to the dissociation of ADP upon hydrolysis. Moreover, the binding of three to four KDLs to MsbA enhances the selectivity in binding ATP over ADP, even after significant hydrolysis and accumulation of ADP. This leads us to propose that the number of KDL molecules binding to MsbA is positively correlated with ATP binding affinity. However, it is essential to note that triggering and completing the transport cycle does not always necessitate the presence of 3 or more KDL molecules. This observation suggests that KDL may exhibit a preference between the interior[10,13] and exterior binding sites[11,17].

Native MS studies provide insightful information to guide structural biology studies. Here, we report several OIF structures. We also report a higher resolution structure of MsbA in an open, OF conformation. This structure is bound to KDL at the exterior site and the NBDs adopt a distinct structure that is not bound to nucleotide. Moreover, tube-like density is observed in the transmembrane region, in which we modeled the C$_{10}$E$_5$ detergent (Supplementary Fig. 16).

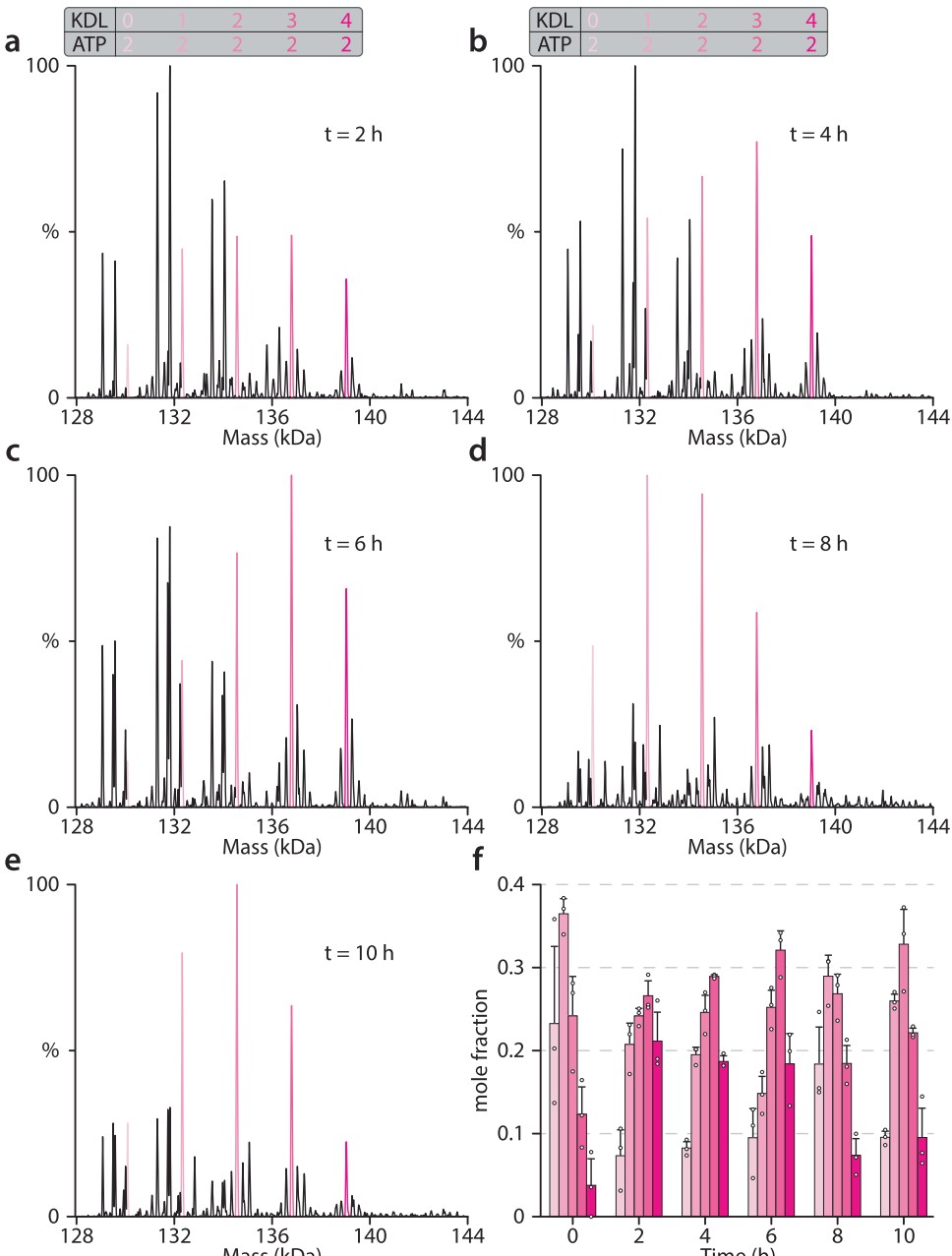

**Fig. 4 | MsbA selectively binds ATP in the presence of KDL at different time points. a–e** Representative deconvoluted mass spectra of 0.5 μM MsbA mixed with 10 μM $Mg^{2+}$, 50 μM ATP and 1 μM KDL are shown. The different incubation times are denoted. The peaks corresponding to MsbA(ATP)$_2$(KDL)$_{0-4}$ are colored in pink. **f** Plot of the mole fraction of MsbA(ATP)$_2$(KDL)$_{0-4}$ at different time points. Reported are the mean and standard deviation ($n = 3$, biological replicates). Source data are provided as a Source Data file.

Interestingly, these densities, one penetrating a hydrophobic pocket between TM5 and TM6 and other nestled between TM1 and TM6, are similar to that observed in our recent structure of MsbA in an open, OF conformation but bound to KDL and ADP-vanadate[11] (Supplementary Fig. 16b, c). It is unclear if lipids bind at these locations and their potential role in regulating MsbA structure. While nucleotide-free structures of MsbA (and other ABC transporters) in IF conformations have been previously reported[21,56,57], the nucleotide-free MsbA in an open, OF conformation reported here is unconventional. However, native MS data does support the existence of nucleotide-free, KDL bound states of MsbA (Fig. 4). Moreover, ATPase activity assays show the transporter is active and ATP is still present under the conditions used for cryoEM (Supplementary Fig. 12). It is also important to note

that ADP accumulates as MsbA turnovers ATP. Despite the higher affinity for ADP binding to MsbA, neither ADP nor ATP is bound in the NBDs. Nevertheless, additional work is needed to unequivocally establish the nucleotide-free open, OF structure in the context of the MsbA transport cycle.

Taken together with other reported structures, a hypothetical model of the MsbA transport cycle can be rendered (Fig. 7). Starting from OIF4, the most open structure, the NBDs come together (Fig. 7a–d), likely as a result of LOS and ATP binding. Through the power of ATP hydrolysis, MsbA adopts an open, outward conformation (Fig. 7e–h), resulting in flipping of LOS from the cytoplasmic to periplasmic leaflet of the inner membrane. However, vanadate-trapped MsbA can adopt two OF conformations: open and bound to

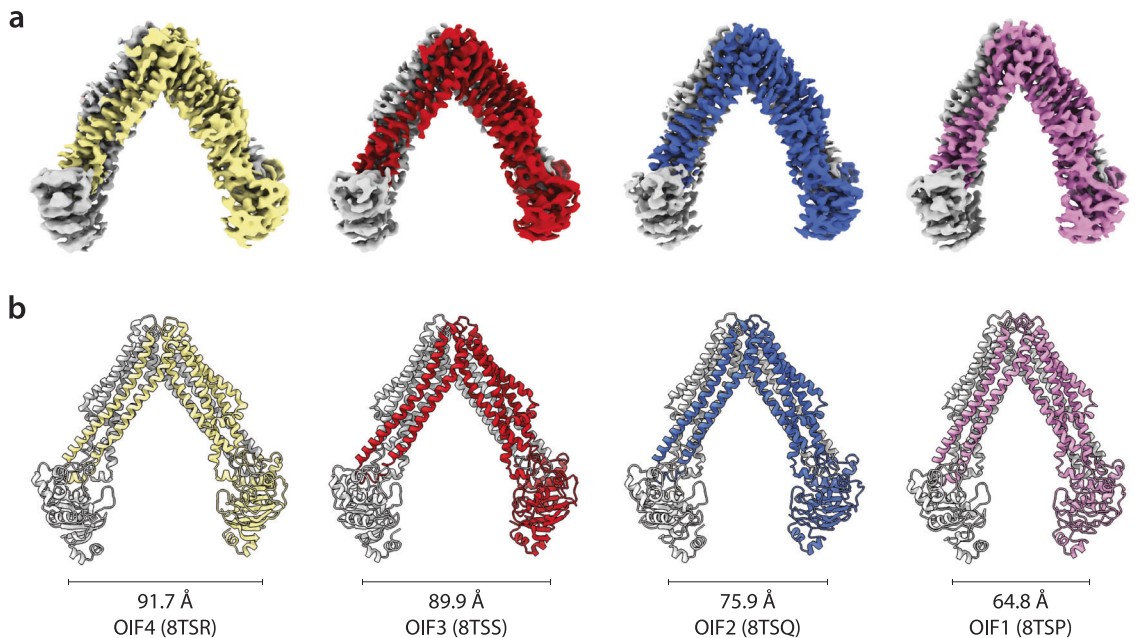

**Fig. 5 | Different open, inward-facing MsbA structures. a** CryoEM density map colored by subunit. **b** The different structures are shown in cartoon representation and colored by subunit. The distance between NBDs (T561 Cα to Cα) is shown.

91.7 Å
OIF4 (8TSR)

89.9 Å
OIF3 (8TSS)

75.9 Å
OIF2 (8TSQ)

64.8 Å
OIF1 (8TSP)

KDLs at the exterior site (Fig. 7f); and occluded (Fig. 7g). It remains unclear what determines the open versus occluded OF conformations of MsbA, and if KDL binding is required to adopt the open state. It is often thought that post hydrolysis MsbA remains bound to ADP. The nucleotide-free open, outward facing conformation presented here suggests nucleotide is released post ATP hydrolysis prior to repopulating an IF conformation (Fig. 7h and Supplementary Movie 1). Taken together, these structures provide additional insight into the MsbA transport cycle.

## Methods

### Protein expression and purification

The expression and purification of *E. coli* MsbA (Uniprot P60752) has been previously described[11]. In brief, the expression plasmid for MsbA with an N-terminal TEV protease cleavable His$_6$ tag was transformed into *E. coli* BL21-AI competent cells (Invitrogen). A single colony was picked and grown overnight in LB Broth at 37 °C. The overnight culture was used to inoculate Terrific Broth and grown at 37 °C until the OD$_{600nm}$ reached a value between 0.6 and 1.0. Protein expression was induced by the addition of 0.5 mM isopropyl β-D-1-thiogalactopyranoside (IPTG) and 0.2% (w/v) arabinose and grown overnight at 25 °C. The cells were harvested by centrifugation at 5000 $g$ for 10 min, washed once with lysis buffer (30 mM TRIS, 300 mM NaCl, pH 7.4) and re-pelleted. Pellets were stored at −80 °C prior to use.

Cell pellets were thawed and resuspended in lysis buffer and lysed by four passages through a M-110P Microfluidizer (Microfluidics) on ice operating at 25,000 psi. The lysate was centrifuged at 20,000 $g$ for 25 min at 4 °C. The resulting supernatant was centrifuged at 100,000 $g$ for 2 h at 4 °C to pellet membranes. The pelleted membranes were resuspended and homogenized in membrane resuspension buffer (30 mM TRIS, 150 mM NaCl, 20% (v/v) glycerol, pH 7.4). Membrane proteins were extracted with 1% (w/v) DDM and stirred overnight at 4 °C. The extraction was clarified by centrifugation (40,000 $g$, 20 min) and filtered using a 0.45 μm syringe filter. The clarified lysate was supplemented with 10 mM imidazole and 1 mM MgCl$_2$ prior to loading onto a Ni-NTA (Qiagen) column equilibrated with NHA buffer (20 mM TRIS, 150 mM NaCl, 10 mM imidazole, 1 mM MgCl$_2$, 10% (v/v) glycerol, pH 7.4) supplemented with 0.02% DDM. The column was then washed with 5 column volumes (CV) of NHA buffer. The bound protein was then treated with 10 CV of NHA buffer supplemented with 2% (w/v) NG. The column was then re-equilibrated with 7 CV of NHA buffer and eluted with 3 CV of NHA buffer supplemented with 500 mM imidazole. The eluted protein was loaded onto a HiPrep 26/10 desalting column (GE Healthcare) equilibrated with 20 mM TRIS, 150 mM NaCl, 10% (v/v) glycerol, pH 7.4. Peak fractions containing the desalted membrane protein were collected. The pooled sample was then treated with TEV protease (produced in-house)[58] overnight at room temperature to cleave the N-terminal His$_6$ tag. The digested material was passed over Ni-NTA beads equilibrated with NHA buffer and the flow-through containing the tag-less protein was collected. Another 5 CV of NHA wash was then applied and collected as well. The collected material was concentrated using a centrifugal concentrator (Millipore, 100 kDa molecular weight cutoff) followed by injection onto a Superdex 200 Increase 10/300 GL (GE Healthcare) column equilibrated in GF buffer (20 mM TRIS, 150 mM NaCl, 10% (v/v) glycerol, 0.065% C$_{10}$E$_5$). Peak fractions containing dimeric MsbA were pooled, concentrated, flash-frozen, and stored at −80 °C prior to use.

### Preparation of MsbA samples for native MS

To saturate the N-terminal copper(II) binding site of MsbA, copper(II) acetate was added to a final concentration of 20 μM prior to buffer exchange into 200 mM ammonium acetate supplemented with 0.065% C$_{10}$E$_5$ using a Micro Bio-Spin 6 (BioRad) desalting column, which also removes excess copper(II). Trisodium ATP was dissolved in water and the pH was adjusted to 7 using 1 M NaOH. To remove sodium ions, the dissolved ATP was injected into a HiTrap Q HP column (Cytiva Life Sciences) equilibrated in H$_2$O and the bound ATP was eluted with 35% of 2 M ammonium acetate. The peak containing ATP was pooled and concentration was determined using absorbance at 260 nm with an extinction coefficient of 15400 M$^{-1}$ cm$^{-1}$.

### Native mass spectrometry

Samples were loaded into gold-coated borosilicate glass capillaries (prepared in-house)[32] and were ionized via nano electrospray into an Exactive Plus EMR Orbitrap Mass Spectrometer (Thermo Scientific). For native mass analysis, the instrument was tuned as follows: spray voltage 1.70 kV, capillary temperature 200 °C, collision-induced

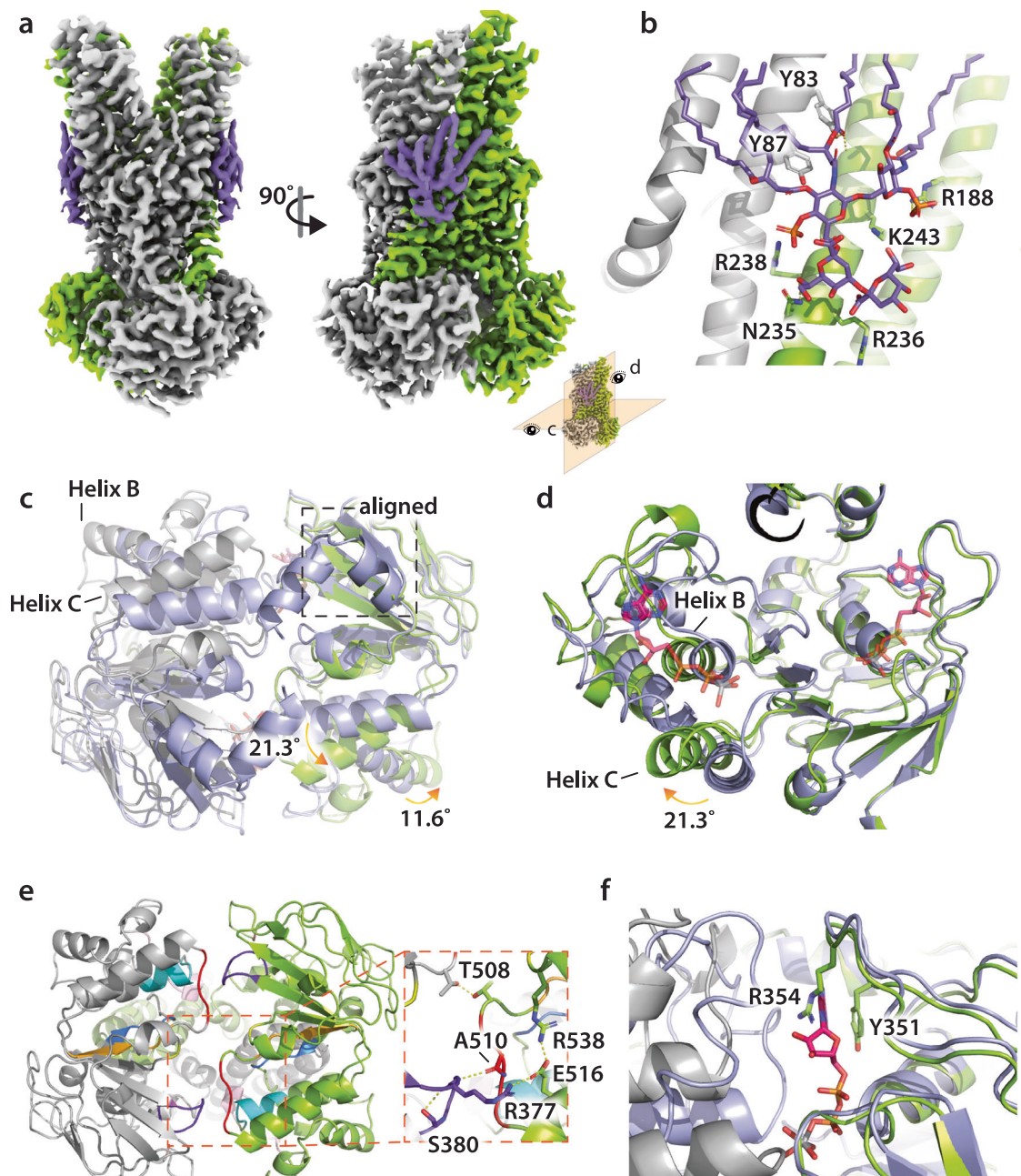

**Fig. 6 | Structure of nucleotide-free and KDL bound to MsbA in an open, outward-facing conformation. a** The 2.7 Å cryoEM density map colored by sub-unit. The density for KDL is colored purple. **b** View of KDL bound to the exterior site shown in stick representation. **c, d** Different views of the NBDs shown in cartoon representation. The vanadate-trapped MsbA structure (PDB 8DMM) is aligned to one chain and colored light blue. The ADP and vanadate bound in PDB 8DMM are shown in stick representation. **e** Key interactions stabilizing the NBD interface. Conserved NBD motifs are shown with the A-loop in salmon, Walker A (P-loop) in purple-blue, Q-loop in marine, X-loop in light pink, C-loop in cyan, Walker B in bright orange, D-loop in red, and the H-switch in yellow. **f** The π–π interaction observed in nucleotide-bound structures is substituted by a cation-π interaction.

dissociation (CID) 65.0 eV, collision energy (CE) 100, trapping gas pressure 6.0, source DC offset 10 V, injection flatapole DC 8.0 V, inter flatapole lens 4 V, bent flatapole DC 3 V, transfer multipole DC 3 V, C-trap entrance lens 0. Mass spectra were acquired with resolution of 17,500, microscans of 1 and averaging of 100.

Obtained mass spectra were deconvoluted using UniDec[52]. Peak intensities for apo and nucleotide-bound protein were determined and converted to mole fractions to compute the relative abundance for each species for each independent experiment. For MsbA (P) binding the $n$th ligand ($L_n$), we applied the following sequential ligand binding model:

$$PL_{n-1} + L \iff^{K_{An}} PL_n \quad K_{An} = \frac{[PL_n]}{[PL_{n-1}][L]} \quad (1)$$

$[P]_{total}$ represents total protein concentration:

$$[P]_{total} = [P] + \Sigma_{i=1}^{n}[PL_i] = [P] + \Sigma_{i=1}^{n}[P][L]^i \Pi_{j=1}^{i} K_{Aj} \quad (2)$$

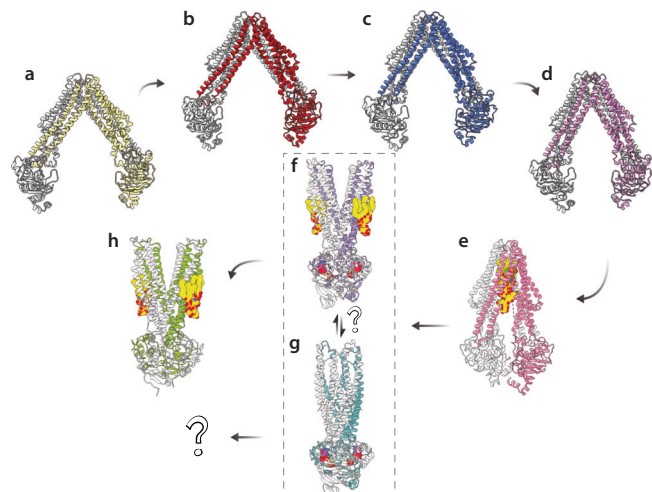

**Fig. 7 | Snapshots of the MsbA transport cycle. a–d** MsbA can adopt open, inward-facing conformations that are dynamic and vary in their degree of openness. Shown are PDBs (**a**) 8TSR (**b**) 8TSS (**c**) 8TSQ and (**d**) 8TSP. **e** Upon binding LPS and nucleotide, MsbA forms a closed, inward-facing conformation. Shown is MsbA bound to LPS and inhibitor G907 (PDB 6BPL). **f, g** The transition state of MsbA can be trapped with vanadate and ADP. Shown are outward-facing conformations in (**f**) open (PDB 8DMM) and (**g**) occluded (PDB 7BCW) states. It remains unclear if these conformations are in equilibrium and if the binding of KDL is required to shift the equilibrium to the open state. **h** After ATP hydrolysis, MsbA maintains an open, outward-facing conformation but the NBDs structure changes, resulting in the release of nucleotide. MsbA then repopulates an open, inward-facing conformation to restart the transport cycle. KDL or LPS is shown in yellow. ADP is shown in magenta.

The above equation can be rearranged to calculate the mole fraction ($F_n$) of $PL_n$:

$$F_n = \frac{[PL_n]}{[P]_{total}} = \frac{[L]_{free}^n \prod_{j=1}^n K_{Aj}}{1 + \sum_{i=1}^n [L]_{free}^i \prod_{j=1}^i K_{Aj}} \quad (3)$$

where $[L]_{free}$ is the free ligand concentration at equilibrium, which can be calculated with known $[P]_{total}$:

$$[L]_{free} = [L]_{total} - [P]_{total} \sum_{i=1}^n i F_i \quad (4)$$

To obtain $K_{An}$, the sequential ligand binding model was globally fit to the mole fraction data by minimization of pseudo-$\chi^2$ function:

$$\chi^2 = \sum_{i=0}^n \sum_{j=1}^d (F_{i,j,\exp} - F_{i,j,calc})^2 \quad (5)$$

where $n$ is the number of bound ligands and $d$ is the number of the experimental mole fraction data points.

## Sample preparation for single-particle cryoEM

To prepare samples for cryoEM studies, MsbA was pre-saturated with copper(II). Excess copper and glycerol were removed using a desalting column. Peak fractions were pooled and concentrated to 10 mg ml$^{-1}$. Vitrification was performed using a Vitrobot Mark IV (Thermo Fisher) operating at 8 °C and 100% humidity. A total of 3.5 μL of sample in cryoEM buffer (150 mM NaCl, 20 mM TRIS, 0.065% $C_{10}E_5$, pH 7.4) incubated with 1 mM MgCl$_2$, 1 mM ATP and 194 μM KDL at 4 °C for 6 h was applied to holey carbon grids (Quantifoil 300 mesh Cu 1.2/1.3) glow-discharged for 30 s. The grids were blotted for 5 s at blotting force 1 using standard Vitrobot filter paper (Ted Pella, 47000-100), and then plunged into liquid ethane.

## Data collection for single-particle cryoEM

Data collection was performed at the Advanced Electron Microscopy Facility at the University of Chicago. The dataset was collected as movie stacks with a Titan Krios electron microscope operating at 300 kV, equipped with a K3 direct detector camera. Images were recorded at a nominal magnification of 81,000× at super-resolution counting mode by image shift. The total exposure time was set to 4 s with a frame recorded every 0.1 s, resulting in 40 frames in a single stack with a total exposure around 50 electrons/Å$^2$. The defocus range was set at −1.0 to −2.5 μm. See Supplementary Table 4 for the details of data collection parameters.

## Image processing for single-particle cryoEM

Collected movies were processed using CryoSPARC[59] and RELION[60]. The detailed data processing flow is shown in Supplementary Fig. 13. Briefly, stage drift and anisotropic motion of the stack images were corrected by patch-based motion correction. CTF parameters for each micrograph were determined by patch-based CTF estimation. Blob picker followed by template picker were used for particle picking. The particles were cleaned by two rounds of 2D classification. Four initial models were generated from the remaining particles using ab initio reconstruction. Two major conformations corresponding to outward-facing and inward-facing structures were identified. The outward-facing and inward-facing particles were exported to RELION separately for further processing, including 3D classification, CTF refinement and polishing. The inward-facing particles were further classified into four conformations (OIF1-4). After polishing in RELION, the particles were imported back to CryoSPARC for non-uniform refinement with per-particle defocus and CTF optimization. The outward-facing class resulted in a final map of 2.7 Å. The inward-facing classes were individually refined, resulting in final maps ranging from 3.6 to 3.9 Å resolutions (Supplementary Table 4). Both C1 and C2 symmetry were tested for the reconstructions with the latter yielding slightly better maps.

## Model building, refinement, and validation for single-particle cryoEM structures

For open, inward-facing structures, the previously reported structure of MsbA (PDB 8DMO)[11] was docked into the cryoEM maps using Chimera[61]. The model was manually refined using Coot[62]. A similar approach was used for the open, outward-facing conformation but PDB 8DMM was used. The final models underwent multiple rounds of real-space refinement using Phenix[63]. Coot was used to manually fix geometry outliers. Figures were generated using ChimeraX[64] and Pymol (Schrödinger LLC., version 2.1). See Supplementary Table 5 for the details of model statistics.

## Reporting summary

Further information on research design is available in the Nature Portfolio Reporting Summary linked to this article.

## Data availability

MsbA cryoEM structures and maps have been deposited in the PDB and EMDB as follows: 8TSO and EMD-41596, 8TSP and EMD-41597, 8TSQ and EMD-41598, 8TSS and EMD-41560; and 8TSR and EMD-41599. Previously reported protein structures used in this study are: 3B5W (open, inward-facing MsbA), 8DMO (open, inward-facing MsbA), 6BL6 (open, inward-facing *Salmonella typhimurium* MsbA), 8DMM (vanadate-trapped MsbA bound to KDL), 6BPL (MsbA in complex with LPS and G907), and 7BCW (vanadate-trapped MsbA). Native MS data has been deposited at Zenodo (https://doi.org/10.5281/zenodo.10845033). Source data are provided with this paper.

## Code availability

Python code to determine individual equilibrium binding constants is available at https://github.com/LaganowskyLab (https://doi.org/10.5281/zenodo.11040823).

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

## Acknowledgements

This work was supported by National Institutes of Health (NIH) under grant numbers (R01GM121751, R01GM139876, R01GM138863, and RM1GM145416 to A.L.; and R35GM143052 to M.Z.). We thank the staff at the University of Chicago Advanced Electron Microscopy (RRID: SCR_019198) for the help with cryo-EM data collection. We thank the Research Computing Center at the University of Chicago for the support of this work by providing the computing resources of the Beagle3 HPC cluster funded by NIH (S10OD028655).

## Author contributions

T.Z. and A.L. designed the research. T.Z. and J.L. expressed and purified MsbA. S.Y. and E.S. helped prepare MS samples. T.Z. performed mass spectrometry experiments. T.Z. and A.L. analyzed the data. B.Y., M.Z., and A.L. collected and processed cryo-EM data and built and refined atomic models. T.Z. and A.L. wrote the manuscript with input from the other authors.

## Competing interests

The authors declare no competing interests.
