## [Peer Review File · Nature Communications]

Native mass spectrometry and structural studies reveal modulation of MsbA-nucleotide interactions by lipidsREVIEWER COMMENTS

Reviewer #1 (Remarks to the Author):

MsbA is a member of the ATP-binding cassette (ABC) superfamily of integral membrane proteins and plays an essential role in LPS biosynthesis in *E. coli* by facilitating the flipping of the LPS-precursor lipooligosaccharide (LOS) from the cytoplasmic side of the inner membrane to the periplasmic side. This study defines the role of individual nucleotide (ATP & ADP) binding to various states of the turnover cycle using native mass spectrometry combined with cryo-EM to construct the most detailed structural model of LOS translocation to date.

The work is of considerable interest to all those interested in directional transport of lipids across biological membranes.

The mass spectrometry work is quite beautiful; the ability to perform a native MS experiment on a twelve-membrane spanning integral membrane protein and resolve bound lipids and single nucleotides, ADP versus ATP, is a technical marvel. But it is the combination with structural biology that elevates the work to the very highest standard. The manuscript is very well prepared and I found no typos. The data is excellent and presented accurately.

The KD values for ATP and ADP are determined for each molecule bound and reflect considerably tighter binding than reported previously. The effect of bound lipids can be clearly seen.

The temperature of the experiments should be stated and if not performed at 37C the reason should be justified. I can't see what temperature was used for Figure 4, for example.

Reviewer #2 (Remarks to the Author):

In their manuscript, Zhang et al. utilize native mass spectrometry to investigate how lipids affect nucleotide binding in MsbA, one of the most studied bacterial ABC transporters and lipid flipase. The MS study is supplemented by cryo-EM analysis, which is well performed, despite the highly unexpected results. The study is undoubtedly interesting and I am quite fascinated about the power of native MS in this specific application, however, I have a number of concerns in terms of experimental design and interpretation that need to be addressed.

The authors should explain why they are using excess of ATP over Mg²⁺ in their measurements; typically, it is other way around. In such experimental setup, they end up having a mixture of MgATP and ATP, which have different charges and different binding kinetics. The same applies to ADP and Mg²⁺. How does this affect determination of equilibrium dissociation constants? The authors should determine the Kds for ATP/ADP, using Mg²⁺ and ATP/ADP in equimolar ratios.

For characterization of lipid and ADP binding to MsbA, the authors used 25 μM ADP and 10 μM Mg^{2+} , for characterization of lipid and ATP binding – 50 μM ATP and 10 μM Mg^{2+} , while deconvoluted mass spectra with nucleotides only were displayed for 20 μM ATP/ADP and 10 μM Mg^{2+} . Why did the authors choose different ratios of ATP/ADP over Mg^{2+} for different experiments?

What is the mass resolution in these measurements? Can the authors discriminate between ATP and MgATP in their spectra? They should discuss this.

To make Figure 1b and 1d clearer, x-axes should be sampled finer (0.5 kDa instead of 2 kDa). Additionally, the authors should add Supplementary table(s) with all theoretical and experimentally obtained masses.

My main concern is regarding the interpretation of the highly unexpected OF open conformation with separated NBDs, which is quite interesting, but most likely artificial. Transporters cannot have both extracellular and intracellular gates open at the same time. I agree with the authors that it is not an artifact from cryo-EM specimen preparation, but could still be an artifact of purification. A number of detergent molecules are tightly bound in the OF structure, and some of them are extending towards the space between the TMDs, which could prevent the extracellular gate from closing and lock the transporter in such artificial conformation. The authors should discuss this. The lack of bound nucleotides in the OF structure under given conditions additionally indicates problems with sample preparation. Was this particular preparation active? What is generally the ATPase activity of MsbA in C10E5 compared to commonly used detergents, like DDM or LMNG? The authors should perform activity assay on this preparation, if this is not possible anymore, repeat cryo-EM experiment on a new prep, after confirming its activity. How can one even know that the structures are really under turnover and all the ATP hasn't been consumed yet if the ATPase activity assay is not shown? In any case, I do think that it is still worth putting such a structure out to make community aware about the artefacts that can appear during sample preparation, but the way the structure is presented and discussed should be different. For publication in such a journal, it is not sufficient just to put out the conformation, which is contradicting all the current knowledge about transporters, and not discuss this issue at all. In line with this, such artificial conformation cannot be included in the transport cycle of MsbA (Figure 7) unless proven otherwise by additional methods.

We thank the reviewers for their time and critique of our manuscript.

REVIEWER COMMENTS

Reviewer #1 (Remarks to the Author):

MsbA is a member of the ATP-binding cassette (ABC) superfamily of integral membrane proteins and plays an essential role in LPS biosynthesis in *E. coli* by facilitating the flipping of the LPS-precursor lipooligosaccharide (LOS) from the cytoplasmic side of the inner membrane to the periplasmic side. This study defines the role of individual nucleotide (ATP & ADP) binding to various states of the turnover cycle using native mass spectrometry combined with cryo-EM to construct the most detailed structural model of LOS translocation to date. The work is of considerable interest to all those interested in directional transport of lipids across biological membranes.

The mass spectrometry work is quite beautiful; the ability to perform a native MS experiment on a twelve-membrane spanning integral membrane protein and resolve bound lipids and single nucleotides, ADP versus ATP, is a technical marvel. But it is the combination with structural biology that elevates the work to the very highest standard. The manuscript is very well prepared and I found no typos. The data is excellent and presented accurately.

The KD values for ATP and ADP are determined for each molecule bound and reflect considerably tighter binding than reported previously. The effect of bound lipids can be clearly seen. The temperature of the experiments should be stated and if not performed at 37C the reason should be justified. I can't see what temperature was used for Figure 4, for example.

We incubated samples on ice to slow the reaction. The manuscript has been revised to make note of this point.

Reviewer #2 (Remarks to the Author):

In their manuscript, Zhang et al. utilize native mass spectrometry to investigate how lipids affect nucleotide binding in MsbA, one of the most studied bacterial ABC transporters and lipid flipase. The MS study is supplemented by cryo-EM analysis, which is well performed, despite the highly unexpected results. The study is undoubtedly interesting and I am quite fascinated about the power of native MS in this specific application, however, I have a number of concerns in terms of experimental design and interpretation that need to be addressed.

The authors should explain why they are using excess of ATP over Mg²⁺ in their measurements; typically, it is other way around. In such experimental setup, they end up having a mixture of MgATP and ATP, which have different charges and different binding kinetics. The same applies to ADP and Mg²⁺. How does this affect determination of equilibrium dissociation constants? The authors should determine the Kds for ATP/ADP, using Mg²⁺ and ATP/ADP in equimolar ratios.

Excellent point. We initially opted for keeping the Mg²⁺ concentration at low levels. The primary reason was to avoid adduction of the cation, which can hinder the quality of mass spectra (for example see **Supplementary Figure 2d** in the revised manuscript). As suggested, we have reacquired ADP and ATP binding data in the presence of excess (50 μM) Mg²⁺, which is shown in **Supplementary Figure 3**. No statistical difference was found for the equilibrium binding constants at different Mg²⁺ concentrations (10 vs 50 μM see **Supplementary Table 2**). Even at higher Mg²⁺ concentrations we observe no appreciable difference in nucleotide binding. However, we do find that higher Mg²⁺ enhances the ATP hydrolysis rate. The revised manuscript includes this new data.

For characterization of lipid and ADP binding to MsbA, the authors used 25 μM ADP and 10 μM Mg^{2+} , for characterization of lipid and ATP binding – 50 μM ATP and 10 μM Mg^{2+} , while deconvoluted mass spectra with nucleotides only were displayed for 20 μM ATP/ADP and 10 μM Mg^{2+} . Why did the authors choose different ratios of ATP/ADP over Mg^{2+} for different experiments?

As the new results show no difference in nucleotide binding constants in the presence of Mg^{2+} at higher concentration (see our response above), we did not repeat these experiments. Below is a mass spectrum for 0.5 μM MsbA containing 50 μM Mg^{2+} , 50 μM ATP and 1 μM KDL. The only difference between this and that shown in Fig. 4 is the Mg^{2+} concentration. In short, the native mass spectra are comparable for the different concentrations of Mg^{2+} .

What is the mass resolution in these measurements? Can the authors discriminate between ATP and MgATP in their spectra? They should discuss this.

Unfortunately, the resolution of the measurements is not sufficient to discriminate between ATP and MgATP . As the charge states are around 22^+ , the difference between Mg and MgATP in m/z is ~ 1 .

To make Figure 1b and 1d clearer, x-axes should be sampled finer (0.5 kDa instead of 2 kDa). Additionally, the authors should add Supplementary table(s) with all theoretical and experimentally obtained masses.

Figure 1b and 1d. have been revised as suggested. **Supplementary Table 1** has been added with theoretical and experimental masses.

My main concern is regarding the interpretation of the highly unexpected OF open conformation with separated NBDs, which is quite interesting, but most likely artificial. Transporters cannot have both extracellular and intracellular gates open at the same time. I agree with the authors that it is not an artifact from cryo-EM specimen preparation, but could still be an artifact of purification. A number of detergent molecules are tightly bound in the OF structure, and some of them are extending towards the space between the TMDs, which could prevent the extracellular gate from closing and lock the transporter in such artificial conformation. The authors should discuss this. The lack of bound nucleotides in the OF structure under given conditions additionally indicates problems with sample preparation. Was this particular preparation active?

Yes, the quality of the sample was assessed by native MS and was active before freezing samples for cryo-EM.

What is generally the ATPase activity of MsbA in C10E5 compared to commonly used detergents, like DDM or LMNG? The authors should perform activity assay on this preparation, if this is not possible anymore, repeat cryo-EM experiment on a new prep, after confirming its activity.

We have performed these experiments as suggested (see below). The results show MsbA has similar ATPase activity in DDM and C10E5. In addition, MsbA activity in both detergents is stimulated by the addition of KDL.

How can one even know that the structures are really under turnover and all the ATP hasn't been consumed yet if the ATPase activity assay is not shown?

We agree with the reviewer. To test, we have prepared a sample under the same conditions used for cryo-EM studies. At the same concentration of MsbA, ATP, and Mg⁺, we find the transporter is still active, turning over ATP even 6 hours after the time point we froze (see **Supplementary Figure 12**). The result clearly shows that there is still ATP left. The revised manuscript has been updated accordingly.

In any case, I do think that it is still worth putting such a structure out to make community aware about the artefacts that can appear during sample preparation, but the way the structure is presented and discussed should be different. For publication in such a journal, it is not sufficient just to put out the conformation, which is contradicting all the current knowledge about transporters, and not discuss this issue at all. In line with this, such artificial conformation cannot be included in the transport cycle of MsbA (Figure 7) unless proven otherwise by additional methods.

The reviewer has failed to recognize that native MS data does support the existence of nucleotide-free, KDL bound states of MsbA (see **Fig. 4a-c**). In addition, the new data shows MsbA is clearly active, stimulated by KDL (as expected), and ATP is still present under conditions used for cryoEM (see **Supplementary Figure 12**). We have added to the discussion regarding this point and make note that ADP accumulates. Despite the higher affinity for ADP binding to MsbA, neither ADP nor ATP is bound in the NBDs. The new MsbA structure (at a resolution of 2.7Å) is unique and exciting, providing an additional snapshot into the transport cycle of MsbA.

REVIEWER COMMENTS

Reviewer #1 (Remarks to the Author):

The author has responded to all reviewer's comments. I believe its ready for publication.

Reviewer #2 (Remarks to the Author):

While the authors did improve the manuscript by adding activity data and testing ADP and ATP binding in the excess of Mg^{2+} , unfortunately, they failed to address my main concern about the interpretation of the unconventional OF-IF structure. I do believe that the structure is correct, but since such conformation and unexpected partial opening of the NBDs have never been observed for any of ABC transporters before, it requires proper discussion. Its physiological relevance is highly questionable, especially, since the structure is nucleotide free. I didn't fail to recognize the presence of the nucleotide-free, KDL-bound MsbA, as reviewers claim, but MS simply doesn't tell anything about conformations, only about composition and such composition has been already seen in several MsbA structures, but only in IF conformation so far. It may well be that what the authors see is physiologically relevant, but such controversial/dogma-changing results cannot be put out simply as self-explanatory without proper discussion and without putting them in the context of the current literature. In general, I am up for controversies, because they do advance science, but they need to be discussed properly. And that's the only thing that I asked the authors, but they ignored my comments. As just mentioned before, I don't exclude the possibility that the structure might be physiologically relevant, no matter how hard to comprehend this, but it does require strong supporting data and extra work from the authors. I do understand that this would not be an easy task, therefore, I didn't require this, but rather asked the authors to be critical about their data and consider the fact that it can also be an artifact. We often do see various effects from the detergent purification of membrane proteins, for example, the formation of non-physiological, but still active oligomers - something like this could also be the explanation of these findings, which may arise from the purification of MsbA in a rather uncommon detergent (C10E5). As I wrote in my review, in the presented structure several detergent molecules are extending towards the space between the TMDs, which could prevent the extracellular gate from closing and lock the transporter in such artificial conformation, and I simply asked the authors to discuss this. They could have simply written that their structure is very unconventional and currently, they cannot tell whether it is physiologically relevant or not. While they believe in the importance of this conformation, they cannot exclude the fact that this can be the result of purification in this specific detergent. But instead, they decided to ignore my comments and to keep to their bold statement that this conformation is part of the transport cycle without any additional supporting data, which might be misleading, therefore, I cannot accept the manuscript as it is written right now.

Reviewer #3 (Remarks to the Author):

NA, adjudicating reviewer.

We thank the reviewers for their time and critique of our revised manuscript.

REVIEWER COMMENTS

Reviewer #2 (Remarks to the Author):

While the authors did improve the manuscript by adding activity data and testing ADP and ATP binding in the excess of Mg²⁺, unfortunately, they failed to address my main concern about the interpretation of the unconventional OF-IF structure. I do believe that the structure is correct, but since such conformation and unexpected partial opening of the NBDs have never been observed for any of ABC transporters before, it requires proper discussion. Its physiological relevance is highly questionable, especially, since the structure is nucleotide free. I didn't fail to recognize the presence of the nucleotide-free, KDL-bound MsbA, as reviewers claim, but MS simply doesn't tell anything about conformations, only about composition and such composition has been already seen in several MsbA structures, but only in IF conformation so far. It may well be that what the authors see is physiologically relevant, but such controversial/dogma-changing results cannot be put out simply as self-explanatory without proper discussion and without putting them in the context of the current literature. In general, I am up for controversies, because they do advance science, but they need to be discussed properly. And that's the only thing that I asked the authors, but they ignored my comments. As just mentioned before, I don't exclude the possibility that the structure might be physiologically relevant, no matter how hard to comprehend this, but it does require strong supporting data and extra work from the authors. I do understand that this would not be an easy task, therefore, I didn't require this, but rather asked the authors to be critical about their data and consider the fact that it can also be an artifact. We often do see various effects from the detergent purification of membrane proteins, for example, the formation of non-physiological, but still active oligomers - something like this could also be the explanation of these findings, which may arise from the purification of MsbA in a rather uncommon detergent (C10E5). As I wrote in my review, in the presented structure several detergent molecules are extending towards the space between the TMDs, which could prevent the extracellular gate from closing and lock the transporter in such artificial conformation, and I simply asked the authors to discuss this. They could have simply written that their structure is very unconventional and currently, they cannot tell whether it is physiologically relevant or not. While they believe in the importance of this conformation, they cannot exclude the fact that this can be the result of purification in this specific detergent. But instead, they decided to ignore my comments and to keep to their bold statement that this conformation is part of the transport cycle without any additional supporting data, which might be misleading, therefore, I cannot accept the manuscript as it is written right now.

We have revised the discussion to now note that the tube-like densities and the role of these molecules (modeled as detergent) is unclear. In addition, we make note that the tube-like densities are identical to our recent open, OF structure of MsbA bound to KDL and ADP-vanadate (see **Supplementary Figure 16**). This suggests the tube-like densities (perhaps detergent) are not trapping an artificial conformation – they are also present in the structure bound to ADP-vanadate (a conventional structure). Specifically, we write *“Moreover, tube-like density is observed in the transmembrane region, in which we modeled the C10E5 detergent (Supplementary Figure 16). Interestingly, these densities, one penetrating a hydrophobic pocket between TM5 and TM6 and other nestled between TM1 and TM6, are similar to that observed in our recent structure of MsbA in an open, OF conformation but bound to KDL and nucleotide and vanadate (Supplementary Figure 16b and 16c). It is unclear if lipids bind at these locations and their potential role in regulating MsbA structure.”*

The discussion of the revised manuscript now reads *“While nucleotide-free structures of MsbA (and other ABC transporters) in IF conformations have been previously reported, the nucleotide-free MsbA in an open, OF conformation reported here is unconventional. However, native MS data does support the existence of*

nucleotide-free, KDL bound states of MsbA (Fig. 4). Moreover, ATPase activity assays show the transporter is active and ATP is still present under the conditions used for cryoEM (Supplementary Figure 12). It is also important to note that ADP accumulates as MsbA turnovers ATP. Despite the higher affinity for ADP binding to MsbA, neither ADP nor ATP is bound in the NBDs. Nevertheless, additional studies are warranted to further characterize the nucleotide-free open, OF structure, especially for the transporter embedded in the membrane.”

In Fig 7, we propose a model given the new and existing structures of MsbA; Simply offering a plausible explanation for these structures in the context of the transport cycle. This model also highlights some key questions regarding the MsbA transport cycle. We have toned our discussion down to now read the unconventional structure “*suggests*” rather than “*indicates*” nucleotide is released post ATP hydrolysis prior to repopulating an IF conformation.

REVIEWERS' COMMENTS

Reviewer #3 (Remarks to the Author):

NA